# The Effect of Preservation Temperature on Liver, Kidney, and Pancreas Tissue ATP in Animal and Preclinical Human Models

**DOI:** 10.3390/jcm8091421

**Published:** 2019-09-09

**Authors:** Maria Irene Bellini, Janice Yiu, Mikhail Nozdrin, Vassilios Papalois

**Affiliations:** 1Renal Transplant Centre, Belfast City Hospital, Belfast BT97AB, UK; 2School of Medicine, University College London, London WC1E 6BT, UK; 3School of Medicine, Imperial College London, London SW72AZ, UK; 4Renal and Transplant Directorate, Imperial College Healthcare NHS Trust, London W120HS, UK; 5Department of Surgery and Cancer, Imperial College London, London SW72AZ, UK

**Keywords:** machine perfusion, organ reconditioning, cell metabolism, transplantation, ischaemia reperfusion injury

## Abstract

The recent advances in machine perfusion (MP) technology involve settings ranging between hypothermic, subnormothermic, and normothermic temperatures. Tissue level adenosine triphosphate (ATP) is a long-established marker of viability and functionality and is universal for all organs. In the midst of a growing number of complex clinical parameters for the quality assessment of graft prior to transplantation, a revisit of ATP may shed light on the underlying reconditioning mechanisms of different perfusion temperatures in the form of restoration of metabolic and energy status. This article aims to review and critically analyse animal and preclinical human studies (discarded grafts) during MP of three abdominal organs (liver, kidney, and pancreas) in which ATP was a primary endpoint. A selective review of recent novel reconditioning approaches relevant to mitigation of graft ischaemia-reperfusion injury via MP and for different perfusion temperatures was also conducted. With a current reiterated interest for oxygenation during MP, a re-introduction of tissue ATP levels may be valuable for graft viability assessment prior to transplantation. Further studies may help delineate the benefits of selective perfusion temperatures on organs viability.

## 1. Introduction

### 1.1. Current State of Abdominal Transplantation (Liver, Kidneys, and Pancreas)

Organ transplantation remains the most effective treatment of end-stage organ failure. The use and optimisation of protocols in different dynamic MP technologies for abdominal organ preservation (liver, kidney, and pancreas) have been extensively studied in the last decades in single or multi-centre randomised controlled trials (RCTs) [1] and different preclinical human and animal models. In order to expand the current donor pool and match the waitlist demand, there has been an increase in organ retrieval from different donors, such as donation after brain death (DBD), donation after circulatory death (DCD), and extended criteria donor (ECD) [2]. In particular, ECD grafts are associated with poorer transplantation outcomes and there is controversy in terms of greater overall economic costs [3]. Namely, a higher proportion of patients may suffer from early graft dysfunction or delayed graft function (DGF), prolonged hospital stay, and increased morbidity in general: Biliary complications in the case of liver [4], acute renal failure after kidney transplant [5], and the need for insulin therapy in the context of pancreas transplantation [6].

The current major preservation modality is static cold storage (SCS) at 4 °C. Even though SCS is logistically simpler, there is evidence of parenchymal damage by extended periods of ischaemia and lack of oxygen with the use of this modality. If the organ is instead preserved with a dynamic preservation, there is the potential to resuscitate it and restore function, such as the recovery of metabolic energetic status in cell mitochondria, e.g., resynthesis of tissue adenosine triphosphate (ATP), reduction in reactive oxygen species (ROS), and inflammatory cytokines resulted from ischaemia-reperfusion injury (IRI). Viability assessment is also a possibility during ex vivo normothermic machine perfusion (NMP) at 37 °C, which also facilitates the surveillance of perfusion parameters, histological and metabolic analysis of biopsy samples [7] for evaluation of the extent of graft tissue injury, prediction of post-transplantation graft function, and rate of survival.

### 1.2. ATP Depletion and Ischaemia-Reperfusion Injury

The insights and understanding behind the underlying molecular mechanisms of IRI are important for identification of suitable biomarkers and development of different novel reconditioning approaches such as small interfering RNA (siRNA), drug delivery for attenuating pro-apoptotic molecules, and downstream signalling pathways. These interventions could be adopted during MP to counteract damage due to IRI and reduction of the risk of post-transplantation DGF, one of the well-known clinical manifestations of IRI in transplanted grafts [8].

Ischaemia, defined as ‘hypoperfusion of tissues’, is referred to as the deficiency in oxygen-carrying arterial blood supply to tissues. A depletion in oxygen can lead to a collapse of the electron transport chain in mitochondria, which is important for oxidative phosphorylation/aerobic respiration for generation of cellular ATP. It also induces anaerobic metabolism and an increase in lactic acid with a decrease in both cytoplasmic pH [9] and antioxidative agents, such as superoxide dismutases, which break down the ROS responsible of tissue damage [10]. The depletion of ATP, which is a major energy-rich phosphate source, can have detrimental effects on intracellular synthetic, catabolic, transcriptional, and transport processes and overall cellular viability. This includes clumping of nuclear chromatin, ribosome detachment impairing protein synthesis, and dysfunction of ATP-driven pumps, such as membrane sodium-potassium pumps (Na^+^/K^+^ ATPases) and reuptake of Ca^2+^ ions by calcium pumps (Ca^2+^ ATPases) on the endoplasmic reticulum and intracellular enzymatic reactions [11]. In normal physiological conditions, the cell gets rid of excess hydrogen ions (H^+^) by the cooperative activity of two ion pumps. Via the Na^+^/K^+^ ATPases and hydrolysis of ATP, sodium ions are pumped out of the cell in exchange for potassium ions (K^+^). The increase of extracellular sodium ion concentration leads to a flow of Na^+^ ions down the electrochemical concentration gradient into the cell via the Na^+^/H^+^ exchanger, resulting in efflux of H^+^ ions from the cell. On the contrary, an ischaemia-induced reduction in pH can lead to the accumulation of cellular Na^+^ ions, Ca^2+^ ions, H^+^ ions, with hyperosmolarity, which contributes to cellular oedema, due to flow of water into the cell. Lactic acid produced from anaerobic glucose metabolism can lead to further disruption of lysozyme membranes. Consequently, hydrolases can be released, which can destroy intracellular structures. Opening of the mitochondrial permeability transition pore (MPTP), in part due to calcium overload, is suggested to occur in parallel to these changes, which further impedes on ATP production and leads to a decrease in the metabolic energetic status of the cell [11]. Importantly, the residual tissue ATP level is closely related to the duration of SCS and an extended period of energy substrate depletion could increase the severity of IRI and subsequent clinical manifestations, e.g., acute kidney injury and early graft dysfunction [12].

The automatic solution to ischaemia was thought to be prompt reperfusion, but paradoxically, as Jennings et al. [13] revealed in canine heart models, the re-establishment of blood supply with provision of oxygen-carrying red blood cells to the ischaemic heart appeared to be associated with a greater extent of necrosis. In fact, a reintroduction of oxygen and its reaction with single electron can initiate the formation of ROS, which drives oxidative stress. Still extensively researched, it was believed that multiple events underpin reperfusion injury, for instance, the generation of free radicals (chemical species with one or more unpaired electrons) such as ROS and a decrease in antioxidative agents due to ischaemic injury can contribute to oxidative stress. Altogether, damage to DNA by ROS, induction of MPTP opening, and oxidative stress can activate downstream pro-inflammatory cytokine cascades, which leads to cell death [11]. The strategic use and benefits of oxygenation is certainly one of the most debated in the current use of MP technology, therefore this article aims to critically review the evidence of the effects of temperature on ATP levels in retrieved organs through systematic search of studies on MP in Medline, EMBASE, Cochrane Library, and Transplant Library. Animal and preclinical models discussed in this article were all subjected to oxygenated machine perfusion, unless otherwise stated.

### 1.3. Preservation Temperature, Metabolism, and Tissue ATP

The picture of abdominal organ transplantation (liver, kidney, and pancreas) is complex: Due to inherent differences in cell types, the resting metabolic activity and tolerance to cold/warm ischaemia of the liver, kidney, and pancreas are different to each other [14] and this needs to be taken into consideration when using ECD organs as a consequence of a broader acceptance criteria. The heterogeneity in donor age, body mass index (BMI), donor starvation due to prolonged hospitalisation in intensive care unit, and inadequate nutritional support and exercise, could be associated with lower than baseline tissue ATP levels and IRI prior to MP preservation and arrival at recipient site.

The current understanding of the relationship of temperature and metabolic rate stemmed from biochemical models, which were derived from ex vivo reactions in test tubes to studies carried out on three-dimensional organ perfusion culture systems. The rationale behind the use of hypothermic MP (HMP) for organ preservation is that by reducing the temperature, cellular metabolism will also decrease, thus reducing the use of oxygen and the rate of depletion in energy substrates, such as ATP. It was proposed that at 4 °C, the rate of metabolism is 10% of that of normal physiological temperature [15]. The van’t Hoff equation in thermodynamics suggested that at hypothermic temperature, e.g., at 4 °C, the rate of chemical reactions of interest will only be 40% of that in organs perfused under normothermic MP (NMP) at 37 °C. Similarly, the Arrhenius relation predicts that as temperature decreases, the thermal excitation of molecules also reduces [16]. The dilemma lies in the possibility that inter- and intra-molecular bonds and structures of proteins and lipids could be altered by hypothermic temperatures, which contributes to ischaemic injuries and decrease in cell viability. Equally, preservation of organs at physiological temperature (37 °C) could accelerate the rate of breakdown of ATP and shift to anaerobic glycolysis due to higher metabolic activity. Thus, there is still a lack of consensus with regards to the most suitable temperature for optimum restoration of metabolic energetic integrity and resuscitation of the organs before transplantation. Figure 1, Figure 2 and Figure 3 represent electron transport during ATP formation in SCS (Figure 1), hypothermic/subnormothermic conditions (Figure 2), and during normothermia (Figure 3).

The recent technological refinements in dynamic oxygenated machine perfusion techniques and perfusate solutions enable further investigations of the beneficial effects of perfusion at different temperatures and different durations: Hypothermia (4–10 °C), subnormothermia (25–34 °C), normothermia (37 °C), and combined approaches, e.g., hypothermic machine perfusion (HMP) with normothermic machine perfusion (NMP) [16,17] and controlled oxygenated rewarming (COR), which involves a gradual increase of temperature to subnormothermia [18].

ATP is a universal energy-rich phosphate found in cells of every human and animal organ. The expression of tissue ATP is considered to be one of the most sensitive marker for ischaemia and an increase in ATP recovery and total adenine nucleotide levels correlate with better functioning and greater viability of grafts before and after transplantation [19]. With ATP loss being a crucial marker for mitochondrial dysfunction at a cellular level and a critical event during IRI [20], there is the need to revisit the effects of different single or combined perfusion temperatures on ATP levels and novel therapeutic approaches to target IRI injuries and organ reconditioning at these temperatures by compiling findings from recent studies in animal and preclinical models of liver, kidney, and pancreas. These studies are fundamental to the development of MP in human clinical trials and will be immensely beneficial in selecting the optimum temperature for the three types of abdominal organs harvested from healthy to a variety of extended criteria marginal donors.

## 2. Liver Preservation

There were considerably more small (rat) and large (porcine) animal and human preclinical studies that looked into the effects of temperatures ranging from hypothermia (4, 10–12 °C), subnormothermia (20–33 °C), COR (20 or 35 °C), normothermia (37 °C), and combined perfusion approaches (HMP and NMP) on tissue ATP in the liver (Table 1, Table 2, Table 3, Table 4, Table 5 and Table 6).

### 2.1. Should a Combined Approach of HMP and NMP Be Adopted for Maximising Resynthesis of Liver Graft ATP?

More than a decade ago, Dutkowski et al. [33] reported the positive effects of oxygenated HMP (or HOPE) at 4 °C on the repair of IRI, possibly through restoration of ATP in HMP perfused rat livers compared to the deleterious effects on ATP by SCS. Similar beneficial effects on hepatic ATP were also reported by de Rougemont et al. in a porcine DCD liver model [34]. Furthermore, in a clinical case-control study (10 dual hypothermic oxygenated MP or dual hypothermic oxygenated machine perfusion (DHOPE) vs. 20 SCS) carried out by van Rijn et al. [35], hepatic ATP was considered as one of the endpoints. Not only did liver tissue ATP increase significantly during MP, but also comparable levels of tissue ATP immediately post-DHOPE and post liver graft reperfusion in recipient patients were reported. Additionally, a significantly lower peak serum alanine aminotransferase (ALT) level and postoperative serum bilirubin concentrations (day 7) were reported in the DHOPE group. The characteristics and outcomes of preclinical human and animal studies involving oxygenated HMP and NMP are shown in Table 1 and Table 4. While HOPE demonstrated potential in the restoration of metabolic energetic status and bile production of human discarded liver grafts, Westerkamp et al. [21] argued that HOPE may not be as effective in the repair of existing hepatobiliary injuries, since there were no differences in the levels of injury markers such as aspartate aminotransferase (AST), ALT, and lactate dehydrogenase (LDH) when compared to the control group (NMP only). In a rat DCD liver model, in which livers were either subjected to 0.5 h or 1 h of warm ischaemia time (WIT), Schlegel et al. [32] reported significantly higher (*p* < 0.0001) ATP levels in the HOPE group, compared to NMP. Apart from the superiority of HOPE over NMP in terms of protection of mitochondrial function and maintenance of ATP levels, the survival rates were also higher in the two WIT (0.5 h or 1 h) groups with corresponding reduction in parenchymal injury. It is important to note that ex vivo NMP (37 °C) was used in both HOPE (4 h) and NMP (4 h) groups for viability assessment, therefore, it was unclear whether this may imply a longer perfusion time at 37 °C for the NMP group. In a study on discarded DCD and DBD human livers [17], the effect on tissue ATP by a combined HMP (2 h) + NMP (4 h) approach was compared against that of NMP-only group. Even though the end-perfusion tissue ATP levels in viable grafts were similar between the HMP+NMP group and the NMP-only group, during the HMP phase of the combined approach group, a median 1.8-fold increase in tissue ATP was observed. This was in line with the findings of Dutkowski et al. [33], de Rougemont et al. [34], Westerkamp et al. [21], and Schlegel et al. [32], further confirming the beneficial effects of HMP on liver tissue ATP restoration. Moreover, in Boteon et al.’s study [17], there were significant differences in ATP levels between viable and non-viable livers treated by NMP only (2.5 vs. 1.1-fold, *p* = 0.05), which strongly supports the role of tissue ATP as a marker of viability and further warranting its inclusion in the decision-making process of whether a liver graft should be discarded. Comparing the durations of HMP in Schlegel et al.’s study [32] in rats and Boteon et al.’s study [17] in discarded human grafts (Table 6), whether a longer period of hypothermia may subject the liver graft to a longer state of metabolic quiescence and, thus, compromising the initial increase in ATP remains an open question. Now, there appears to be a clear consensus of the beneficial effects of oxygenated HMP on resynthesis of ATP in liver grafts subjected to cold and/or warm ischaemia. Referring to findings in Schlegel et al.’s study [32], Dutkowski et al. [36] further explained that for the case of DCD livers, which were already subjected to warm ischaemia damage prior to perfusion, oxygenated HMP may facilitate the resynthesis of ATP from accumulated electron donors such as nicotinamide adenine dinucleotide (NADH) and succinate, whilst reducing the rate of electron transfer and minimizing the risk of escape of electrons from the electron transport chain in mitochondria compared to a higher normothermic temperature.

Thirteen years on, there are still limited RCTs on HMP perfused livers compared to subnormothermic machine perfusion (SNMP) and NMP, with results of three ongoing RCTs pending publication in the near future [37]. There is no clear-cut answer to whether oxygenated HMP or NMP is more preferable for liver preservation, since perfusion temperatures in both modalities demonstrated positive effects on metabolic ATP recovery in animal and preclinical models. Whether a certain temperature will be better for a specific graft subjected to long/short periods of cold ischaemia or long/short periods of warm ischaemia, respectively, is still highly debated, as demonstrated in the example of DCD livers.

### 2.2. Positive Effects of Subnormothermic Perfusion Temperatures (20–30 °C) Liver ATP Levels

The crux of the problem with choosing the optimal perfusion temperature for achieving the maximal benefits in intracellular metabolic status, functionality, and viability of liver grafts lies in weighing up the benefits and shortcomings (e.g., IRI injuries) associated with these temperatures. It was suggested that subnormothermic temperatures may bridge the gap between metabolic retardation during cold hypothermia and hypoxia at ‘hotter’ normal physiological temperatures, enabling the maintenance of an approximate 25% of physiological metabolic levels in the liver graft [37]. In a number of animal and preclinical studies [18,27,28,29,30,38,39] assessing the effects on resynthesis of tissue ATP by SNMP on livers of different extended criteria, DBD or DCD and steatosis were identified (Table 2, Table 3 and Table 4). Additionally, there are studies of a relatively new perfusion technique, controlled oxygenated rewarming (COR) developed by Minor et al. [18] in Essen, Germany. COR involves a slow, step-wise, and progressive increase of perfusion temperature from 4–8 °C to 20 °C [18] or higher (35 °C) [39], which helps the graft to ‘acclimatise’ to an increase in temperature during ex vivo normothermic reperfusion viability assessment (Table 5).

There is evidence of the positive effects of SNMP on liver tissue ATP in small animal models. The increase in tissue ATP was demonstrated in a rat DCD liver models [22,23] with an associated rise in one-month post-transplantation survival rate (SNMP vs. SCS; 83.3% vs. 0%) [22]. Apart from the ability of SNMP to restore livers subjected to warm ischaemia, the limitations of SNMP in rectifying IRI through regenerating ATP levels to baseline level in rat livers subjected to varying periods of cold ischaemia (24, 48, 72, and 120 h) were shown to be 72 h, equivalent to three days of SCS [24]. Taking the survival rates, a decrease in vessel resistance, and ATP recovery into consideration, 48 h of cold storage may be the limit of the protective effects of SNMP. Extrapolating the studies of SNMP in rat models to discarded human liver grafts with varying degrees of injuries, Bruinsma et al. [25] further demonstrated the effects of SNMP on ATP resynthesis and a reduction in clinical hepatic injury markers (LDH, ALT) in DBD and DCD livers. A negative correlation between tissue ATP and ALT was also demonstrated, suggesting that end-perfusion ATP levels could be effective predictors in determining whether a liver graft is sufficiently ‘repaired’ and ‘reconditioned’ prior to transplantation. In a later model by the same group [26], the underlying metabolic effects and reversal of IRI-induced damage by SNMP (21 °C) on different discarded human livers, namely, DBD, steatotic DCD, non-steatotic DCD with extended WIT (>0.5 h) and control DCD, were analysed with the use of metabolomic profiling and the use of transmission electron microscopy (TEM) for identification of possible mitochondrial ultrastructural changes, together with the monitoring of tissue ATP levels. Overall, there was a significant 4.12-fold increase in liver tissue ATP in the SNMP-treated livers. In terms of the absolute increase in ATP levels, these were highest in the DCD group with WIT < 0.5 h, followed by DCD group with extended WIT and lowest in steatotic DCD group. Since ATP synthesis was closely related to the integrity of mitochondria, the group revealed that a greater severity of structural modifications was indicative of injury, such as changes in appearances of cristae membranes, increase in size due to swelling, and deposition of amorphous material within the mitochondria in DCD (WIT > 0.5 h) and steatotic DCD groups. This study not only demonstrated the emerging role of metabolomic profiling for evaluation of the severity of injury of different extended criteria donor grafts, but also further outlined the underlying mechanisms behind SNMP reconditioning.

So far, SNMP of the liver has shown promising results in targeting major factors that predispose the organ to early graft dysfunction, such as steatosis, extended warm and cold ischaemia, or a combination of these factors. Interestingly, the Vairetti group in Italy suggested that absolute tissue ATP levels in rat liver models were significantly higher in livers perfused at 10 °C (HMP) and 20 °C (SNMP) compared to that in livers perfused at 30 °C and 37 °C (NMP) [31] (Table 5). It was proposed that a depletion in ATP might be reflected in the reduction of bile flow in livers perfused at near or normothermic temperatures. A later study by the same group [38] further demonstrated increases in ATP and glycogen levels in rat livers preserved by SNMP at 20 °C vs. SCS.

### 2.3. Controlled Oxygenated Rewarming (COR) on Liver Tissue ATP

In Minor et al.’s porcine study [18], COR (8 °C to 20 °C) was compared to SNMP, HMP, with SCS as a control. Comparable ATP levels were detected in COR and SNMP groups, but these were significantly higher than that in the HMP group (*p* < 0.05). Apart from an increase in bile production and reduction in both AST and ALT levels, the potential reconditioning effects of COR against IRI were further demonstrated in the reduction of expression of pro-inflammatory cytokines, such as tumour necrosis factor-α (TNF-α) and enzymes involved in apoptosis, such as caspase-3. The same group [39] revealed, in a rat DCD liver model, that there were no significant differences in the increase and resynthesis of ATP and reduction in ALT between the COR20 group and COR35 group.

### 2.4. Novel Therapeutic Methods for Targeting IRI and Reconditioning of the Liver during HMP, SNMP, or NMP

In recent years, there was a massive surge in studies that investigated different novel methods for the repair of IRI-related changes and promotion of organ regeneration in liver transplantation, respectively [37]. The following only covers a small picture of the novel modes of treatment against IRI in animal and clinical studies. This ranges from improvements in surgical techniques, targeting downstream targets of peroxisome proliferator-activated receptor-γ (PPAR-γ) in liver [40], to ‘defatting’ steatotic livers [41] and the use of gene therapies, such as RNA interference by siRNA [42]. He et al. [43] presented a case report of an ischaemia-free liver transplantation (IFLT) surgical technique, involving the donation of a fatty liver. This new technique involves the establishment of in situ NMP in the liver during surgical procurement, in which continuous arterial supply to the liver via the coeliac artery was uninterrupted. After NMP for 270 min, the liver graft was transplanted into the recipient. Primary nonfunction, biliary complications were not reported with gradual decrease in AST and ALT throughout one week post-operatively. In mice models, increase in PPAR-γ expression was shown to be protective against cell death. Identification of FAM3A, a target gene of PPAR-γ, was reported to elevate ATP levels and subsequent activation of the Akt signaling pathway, which inhibits apoptosis, mitigating IRI-related damages in liver cells [40].

From previous reports on steatotic livers, there was interest in assessing the abilities of different ‘defatting’ chemical cocktails to stimulate breakdown of lipid droplets, decreasing overall triglyceride concentration in cells and increasing tissue ATP levels in SNMP-treated rat liver models [41].

Apoptosis or cell death is not only a crucial player in organ reperfusion injury, but also an enemy against graft viability and transplant outcome of the liver. A study reported, for the first time this year, the feasibility of introducing siRNA against the Fas receptor in perfusate solutions used during HMP or NMP. Through lipid nanoparticles-mediated transfection and subsequent endocytosis in hepatocytes of end-HMP or end-NMP perfused liver graft during the ischaemic period, there may be a window of opportunity for the repair of graft damage before transplantation [42]. siRNAs are engineered RNA oligonucleotides, and work by hybridizing to the single-stranded mRNAs transcribed from the target gene, in this case, the Fas ligand. This post-transcriptional gene silencing mechanism prevents the Fas ligand from being produced. Therefore, without binding of the Fas ligand to the FAS receptor, activation of downstream apoptotic signalling pathways will be hampered in hepatocytes.

## 3. Kidney Preservation

In kidneys, there were more studies on HMP compared to other perfusion modalities: Three large animal porcine studies, which researched models with characteristics of different extended criteria donors (e.g., DCD, DBD) [44,45,46] and one preclinical study [47] reporting tissue ATP as one of the primary endpoints between 2010 to 2019 were identified (Table 7).

### 3.1. Oxygenated HMP (4 to 10 °C) Could Restore Synthesis of ATP in Kidneys

Buchs et al. [44] compared the reconditioning effects of oxygenated HMP or HOPE at 4 °C vs. SCS on porcine DCD grafts (Maastricht I and II) subjected to varying lengths of WIT (0 or 0.5 h) and CIT (0 or 4 or 8 or 18 h). In this study, the team designed an MP device that was compatible with magnetic resonance, which facilitates real-time, non-invasive, and in-line ^31^P NMR spectroscopy for monitoring ATP resynthesis in the graft during MP. In groups where grafts were only subjected to WIT or CIT (4 or 8 h), respectively, ATP levels were detectable after HOPE, albeit lower than baseline control level. However, it was shown that after an extended period of SCS, equivalent to 18 h of CIT, ATP levels were almost undetectable, despite the application of HOPE. Cold pulsatile perfusion did not result in ATP resynthesis in grafts subjected to both warm and cold ischaemia (4 h), with only precursors ((phosphomonoesther (PME) and inorganic phosphorus (oPi)) being detected. Extending these findings, Buch et al. proposed that immediate HOPE in mobile perfusion machines for WIT-treated DCD grafts could restore the energetic status of the grafts. Thus, this could restrict the use of SCS and maintain graft viability during transport. However, the reanimation of grafts that were subjected to CIT damage by HOPE may be limited to 8 h of cold storage and not beyond that. The limitations of the study were that no data about the rate of DGF and survival were available, since grafts were not surgically transplanted into the animals after these assessments.

The role of ATP as an essential marker for viability of kidneys, a prerequisite for satisfactory transplantation outcome, was further demonstrated in a porcine study utilizing the same NMR imaging method for tissue ATP measurement in DCD kidneys subjected to pulsatile HOPE (4 °C) by Lazeyras et al. [45]. In order to obtain an overall picture of the energetic status of the graft, ATP (alpha-, beta-, gamma-ATP resonances) and precursors (PME, oPi) were all taken into account. Further findings through this non-invasive ATP assessment method included the superiority of perfusion pO2 of 100 kPa over 50 or 20 kPa in resynthesis of ATP. This suggested that the high level of precursors detected may be due to the reduction of cell’s ability to undergo re-phosphorylation of AMP in PME to energy-rich ATP, but this speculation might be inaccurate due to technical obstacles like overlapping resonances. Nevertheless, findings from both studies highlighted the possibility of encompassing ^31^P NMR spectroscopy with clinical MRI in future studies of the effects of preservation conditions on the viability of graft and a tool to the decision-making process of expanding the donor pool, which is currently subjective and heterogeneous, involving visual inspection (e.g., colour, patchiness, physical injuries) and renal resistance.

The importance of ATP tissue measurements as a quantitative index to be added on to the current restricted panel of clinical evaluation criteria of kidneys was also put forward by Kaminski et al. [48] in a porcine study, which revealed that cortical ATP synthesis was significantly restored in WI-injured grafts treated by HOPE at 8 °C compared to SCS (5.8 vs. 0.06 mmol/L, *p* < 0.01). A 90% reduction of tissue ATP was noted upon 1 h of WIT, and interestingly, the renal cortical tissue ATP levels in HOPE-treated, WIT-injured grafts exceeded that in control grafts (3.3 mmol/L tissue) that were not subjected to WIT.

The metabolic protective effects of HOPE at 4 °C on tissue ATP were also demonstrated in discarded human DBD kidneys subjected to >20 h of cold ischaemia by Ravaioli et al. [47] compared to the SCS group, in which the team investigated different oxygenation parameters (hyperbaric and normobaric oxygen). Taking the similarities of kidneys between porcine and humans in terms of size and function into account, ATP resynthesis appears to be possible in grafts challenged by CIT exceeding 8 h, a limit suggested in Buchs et al.’s study [43]. The significance of this study and the three animal models supported the hypothesis that oxygen delivery HMP (4–10 °C) might favour improvements in cellular ATP levels in grafts subjected to IRI due to prolonged CIT and short WIT (0.5 h). The association between ATP levels in organs prior to transplantation and the reduction in risk of DGF were also previously discussed by Wijemars et al. [49].

### 3.2. The Effects of Higher Perfusion Temperatures (>4–10 °C) on Kidney Viability are Currently Unclear

Quantitative changes to ATP in porcine kidneys perfused at normothermic temperature (30 °C) were only reported in one study [45], in which HMP (4 °C) and SCS were also tested as preservation methods (Table 7). Two hours of CIT on these kidneys was associated with an increase in ADP:ATP ratio prior to preservation, and after perfusion, the ADP:ATP ratios were found to be comparable between NMP, HMP, and SCS groups. Contrary to other studies on HMP of kidneys, Kay et al. did not observe significant changes in tissue ATP even in the HMP group.

A recent porcine transplant study by Bhattacharjee et al. [50] reported that a perfusion temperature of 22 °C resulted in significant improvements on IRI injuries in SCS treated kidneys in terms of histology, suppression of expression of IRI related Toll-like receptor signalling molecules (MyD88, NF-kB and HMGB1), and graft functions, such as increase in urinary output and maximal renal blood flow compared to MP at 15 °C and 37 °C. While it was accepted that a hypothermic temperature may be optimal for the protection of metabolic energetic status and viability of kidney grafts, effects of subnormothermia on tissue ATP remains to be established in comparison to other more extensively studied temperatures.

Current novel treatment methods tested in animal models are centred around ameliorating HMP-induced injuries of kidney grafts [51]. A minimally invasive method called rapid sampling microdialysis (rsMD), which was developed in a porcine HMP model [52], may help monitor tissue ATP levels and other metabolic markers during MP in real-time, promoting further investigations into the relationship of MP temperature and renal tissue ATP. Proteolytic enzymes like matrix metalloproteinases (MMPs) are known to be related to the pathophysiology of fibrotic renal diseases. Tissue MMP-2 was, in particular, implicated in patients suffering from antibody-mediated active renal graft vs. host rejection [53]. In a rat DCD kidney model, Moser et al. [54] tested two inhibitory agents on levels MMPs; these include a siRNA against MMP-2 and together with doxycycline. Additionally, high levels of MMP-2 and MMP-9 were identified from perfusates of human kidney grafts, suggesting that an elevation of MMPs might be related to cold hypothermic perfusion. Apart from a decrease in MMPs, a reduction in inflammatory markers such as LDH was also noted. Identification of natural inhibitors, TIMP-1 and TIMP-2 of MMP-9 and MMP-2, respectively, was useful in the study of biomarkers for predicting early and long-term delayed graft function. Paradoxically, a detection and elevation of urinary TIMP-1 and TIMP-2 at day 1 and 3 were correlated with unsatisfactory eGFR at post-operative day 1, 7, and 14. The authors reasoned that this increase might be due to elevation of MMP-9 and MMP-2 in patients who have undergone kidney transplantation [55]. TIMP-2, also a marker of acute kidney injury, was recently shown to be predictive of delayed graft function in patients having received DCD kidney transplantations [56]. Borrowing the idea from Gilloidy et al.’s [42] report on the use of siRNA to protect the end-perfusion liver graft from end-ischaemia, although currently still elusive, the transfection of MMP-2 siRNA during cold hypothermic perfusion of kidney grafts may be useful for prevention of tubular damage and IRI.

## 4. Pancreas Preservation

With the advent of MP technology in the preservation of liver and kidneys, there is growing interest in its application in pancreas transplantation. The major goal is the maintenance of viability of islet cells including insulin producing beta cells in the graft, which is highly susceptible to IRI damage [57]. Existing challenges include the lack of recognised viability markers, and technical obstacles like susceptibility of vascular endothelial lining to MP flow. ATP generation as a means to evaluate cellular metabolism and as part of the functional assessment of the organ were once again highlighted [58]. The characteristics of two studies on HMP and NMP that reported tissue ATP as one of the primary endpoints are shown in Table 8 and Table 9, respectively.

In a pilot study, dual arterial HMP (4–7 °C) was shown to be effective in increasing tissue ATP by 6.8-fold in 10 human discarded DCD pancreata and by 2.6-fold in 10 DBD pancreata. All pancreata were subjected to 3 to 5 h of cold preservation prior to HMP or SCS. It was noted that prior to HMP, ATP level was significantly lower in DCD pancreata compared to DBD pancreata, possibly related to IRI sustained from WI in the DCD group. However, after HMP, ATP level in DCD pancreata was comparable to that of DBD pancreata. The improvements in viability of HMP-treated pancreata were accompanied by absence of oedema, but an increase in amylase, which was also reported in the SCS group. A higher level of amylase was believed to be associated with poorer transplantation outcomes and may mean acinar and parenchymal damage. However, the authors speculated that the increase in amylase levels after both dynamic and static preservation could be related to experimental factors, such as injuries induced by biopsy collections. Among the 20 samples, viable islets were isolated from 2 DCD pancreata [57]. The relationship between tissue ATP and better transplantation outcome was also reported in earlier studies involving a two-layer preservation method of pancreata, rather than MP [60].

Instead of directly measuring pancreata tissue ATP levels, Kumar et al. [59] studied the effects of ex vivo NMP (37 °C) and high- or low-pressure perfusion on ATP Synthetase Complex V, a marker of viability and an enzyme essential for ATP production in a porcine DCD pancreata model. It was found that low-pressure ex vivo NMP treatment of pancreata resulted in the highest levels of ATP Synthetase Complex V, indicating active metabolic ATP production to support ATP-dependent pathways underlying insulin secretion after glucose stimulation. Amylase was also found to be lower in low pressure NMP treated pancreata compared to SCS. As mentioned above, even though a higher amylase is suggestive of a higher risk of organ dysfunction, more studies will be needed to assess the benefits of a lower temperature (4 °C) over a higher temperature as high as 37 °C. The limitation of the study lies in the use of a grading system for measuring the positive immunohistochemical staining of biopsied samples, instead of measuring the tissue ATP concentration by a commercial bio-illuminescence kit used in most animal and preclinical studies apart from NMR spectroscopy. Careful interpretation of these results is necessary, and a further validation of the beneficial effects of normothermic perfusion temperature on ATP resynthesis in donor pancreata will be needed with the use of kits or NMR spectroscopy. However, the two studies strongly suggested that more research and possible clinical trials in MP of pancreata from different types of ECDs are warranted, and changes in tissue ATP could be considered as part of the criteria for determining viability and functionality of pancreata graft.

## 5. Conclusions

This review analysed the studies in the last decade that had tissue ATP in animal and preclinical models as an endpoint (Table 10).

ATP acts as a convenient mean for addressing one of the major challenges in transplantation: What is the effect of temperature on organ metabolism and how does this correlate with post-transplantation outcomes, such as DGF and survival rate? In the liver (Table 11), MPs of all temperatures were shown to increase tissue ATP levels, but there is less agreement over the temperature range that gives rise to the highest absolute increase in end-perfusion tissue ATP.

In the kidneys (Table 12), with fewer published literature on SNMP and NMP to date, HOPE appeared to be most effective in increasing graft ATP resynthesis.

More evidence is needed on the effects of preservation temperature on donor pancreatic ATP levels (Table 13).

In conclusion, tissue ATP is a useful quantitative index for informing clinicians in donor organ evaluation and a marker for the extent of resuscitation of end-ischaemic grafts by pharmacological and genetic interventions.

## Figures and Tables

**Figure 1 jcm-08-01421-f001:**
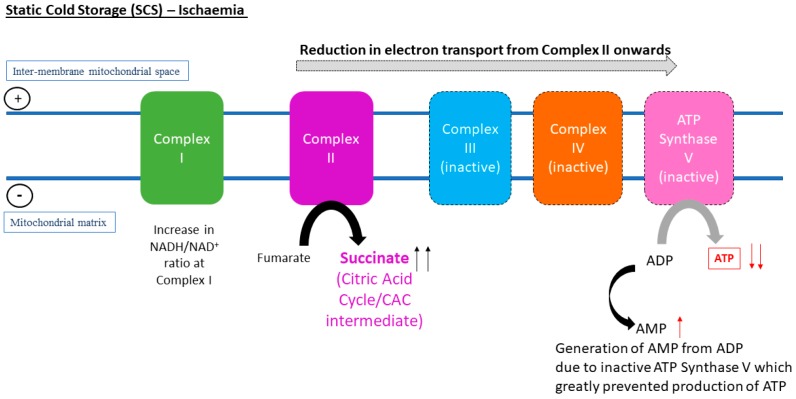
In vivo metabolomic analysis of murine liver and kidneys is associated with a conserved pathway that occurs during the state of ischaemia, which was the accumulation of succinate at Complex II in the electron transport chain.

**Figure 2 jcm-08-01421-f002:**
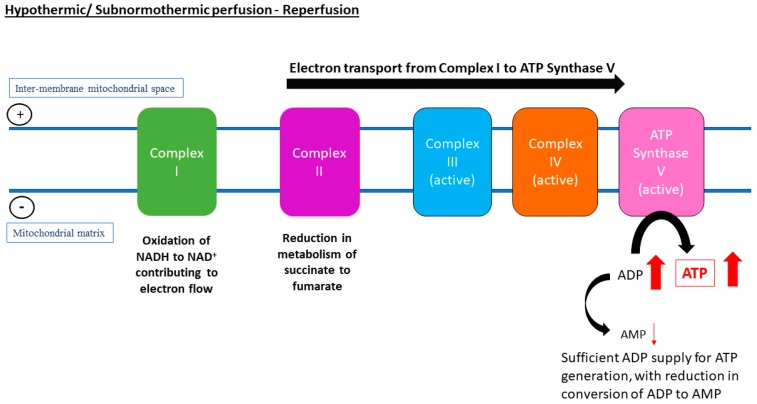
Despite restoration of oxygen supply, a lower temperature during hypothermic machine perfusion (HMP) or subnormothermic machine perfusion (SNMP) of the liver seems to be associated with slower metabolic rates, thus there is a reduction in conversion of succinate to fumarate and reduction in overall electron flow through Complex I to inter-membrane mitochondrial space. Instead, the electron flow through the complexes is uninterrupted, which drives activity of adenosine triphosphate (ATP) synthase for ATP recovery. Other pathways are suggested to help resynthesise tissue adenosine diPhosphate (ADP) and ATP, such as the purine salvage pathway.

**Figure 3 jcm-08-01421-f003:**
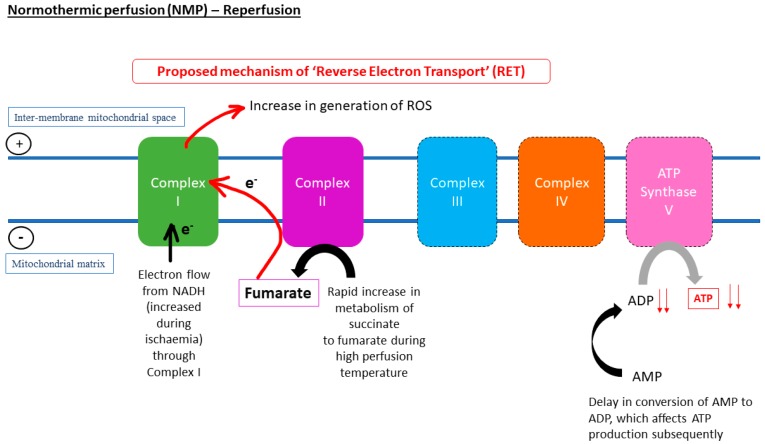
The increase in metabolism of succinate upon restoration of oxygen during reperfusion might lead to reverse electron transport (RET), absence of electron flow to ATP synthase, and delay in conversion of AMP to ATP. Thus, ADP production might lead to a decline in ATP production. Following from this, in the liver, the absence of ADP due to ischaemia and the high temperature during normothermic machine perfusion (NMP) could increase the metabolism of succinate to fumarate, leading to an increase in the flow of electrons through Complex I, which hastened generation of ROS underlying reperfusion injuries.

**Table 1 jcm-08-01421-t001:** Characteristics and outcomes of key pre-clinical and animal study investigating HMP preservation of livers.

Study.	N	WIT, h	CIT, h	Perfusate	Perfusion Time, h	Perfusion Technique *	Temp, °C	Model	End Points	ATP Measurement Technique	Study Outcome
Westerkamp [21]	18	-	7 or 9	Belzer-UW Machine Perfusion solution, pH 7.4, calcium chloride, dextrose, potassium phosphate	2	HMP (ex situ viability testing by NMP)	10–12	Human, discarded livers (DCD with advanced donor age, DBD with high BMI)	Tissue ATP, O_2_ consumption, vascular resistance, bile production, AST, ALT, LDH, GGT, histology	ATP Bioluminescence assay kit CLS II (Roche Diagnostics GmbH, Boehringer Mannheim, Germany)	2 h end-ischaemic HMP preservation was shown to be effective in energy status restoration of DCD liver grafts which showed a >15-fold increase in ATP level and significantly higher bile production versus SCS group. There was no significant reduction in hepatobiliary injury markers. Thus, HMP may be more effective in restoration of energy status and function, instead of reduction of preexisting injury. Further testing in clinical liver transplant trial is suggested.

* oxygenated unless otherwise stated. ALT: Alanine Aminotransferase; AST: Aspartate Transaminase; ATP: Adenosine TriPhosphate; BMI: Body Mass Index; CIT: Cold Ischaemia Time; DBD: Donation after Brain Death; DCD: Donation after Circulatory Death; GGT: gamma-GT; HMP: Hypothermic Machine Perfusion; LDH: Lactate De-Hydrogenase; NMP: Normothermic Machine perfusion; SCS: Static Cold Storage; WIT: Warm Ischaemia Time.

**Table 2 jcm-08-01421-t002:** Characteristics and outcomes of key pre-clinical and animal studies investigating SNMP preservation of livers.

Study	N	WIT, h	CIT, h	Perfusate	Perfusion Time, h	Perfusion Technique *	Temp, °C	Model	End Points	ATP Measurement Technique	Study Outcome
Ferrigno [22]	17	0.5	-	Modified University of Wisconsin –Gluconate solution (modified UW-G) with glucose, adenosine, mannitol	6	SNMP (graft assessed by NMP reperfusion)	20	Rat, DCD livers	Tissue ATP, AST, ALT, LDH, total bile production, GDH, histochemistry	ATP Bioluminescence Assay Kit CLS II (Roche Molecular Biochemicals, Milan, Italy)	An increase in ATP and bile production and a reduction in hepatic damage and GDH levels in NGBD livers were reported in SNMP group vs. SCS group. MP at 20 °C is shown to protect marginal grafts against IRI and graft recovery. A liver transplantation model is needed for further validation.
Berendsen [23]	24	0 or 1	-	WE with sodium bicarb and L-glutamine supplemented with insulin, penicillin, hydrocortisone and heparin	2–3	SNMP	21	Rat, DCD livers	Tissue ATP, ALT, AST, pO_2_, bile production, post-transplantation analysis (blood samples, jaundice, infection, body weight, histology)	ApoSENSOR ATP Luminescence Assay Kit (BioVision Inc, Milpitas, CA, USA)	This study presented a SNMP orthotopic liver transplantation model with oxygenated and supplemented cell culture medium as perfusate. In the two groups subjected to WIT (1 h), SNMP was associated with 83.3% one month survival rate vs. 0% in SCS group. After ~2.5 h of SNMP, ATP levels in WIT group were restored from about 6% of fresh controls to a level exceeding that of fresh controls. Clinical parameters were comparable amongst viable animals. SNMP may have the potential to regenerate DCD livers.
Bruinsma [24]	40	-	0 (fresh) or 24 or 48 or 72 or 120	WE	3	SNMP	20	Rat, livers (procurement after anesthesia)	Tissue ATP, ALT, blood gas analysis, vascular resistance, post-transplantation analysis (blood samples, AST, ALT, total bilirubin, glucose, blood urea nitrogen, albumin, body weight)	Luminescence- based cell viability assay (Biovision, Milpitas, CA)	This study assessed the limits of SNMP to restore liver tissue metabolic energy and viability in grafts subjected to different durations of SCS. ATP level decreased and vascular resistance increased with increased duration of SCS. Compared to non-perfused control groups, increase in ATP levels was reported in all SNMP groups except for 120 h. Cold storage for 72 h appeared to be the limit of SNMP to restore ATP levels to that in fresh liver samples. However, post-transplantation survival rates, ATP recovery and vascular resistance suggested that recoverability of liver viability by SNMP is limited to 48 h of SCS. ATP recovery and vascular resistance are suggested to be useful parameters for assessing the ability of MP to improve liver viability after extended cold preservation.
Bruinsma [25]	7	<1	~11	WE supplemented with insulin, penicillin, streptomycin, hydrocortisone	3	SNMP	21	Human, discarded livers (5 DCD & 2 DBD)	Tissue ATP, ALT, LDH, blood gas analysis, oxygen uptake rate, bile production, ALP, phospholipids, histology	Luminescence-based cell viability assay (BioVision)	An overall 3.7-fold increase in ATP was observed post-SNMP. Clinical parameters, e.g. mean LDH, ALP, ALT, lactate and relative WIT were lower in livers with high ATP levels compared to livers with low ATP levels (not statistically significant). A significant (*p* = 0.02) negative correlation was observed between ATP at the end of SNMP and ALT values (which is a marker of hepatic injury). Ex vivo SNMP was shown to be effective in the maintenance of post-ischaemic liver function with improvement of hepatobiliary parameters and metabolic energy status.
Bruinsma [26]	21	0 or <0.5 or >0.5	-	Nutrient-rich, cell-free, and oxygenated perfusate (exact composition not stated)	3	SNMP	21	Human, discarded livers (DBD, steatotic DCD, non-steatotic DCD with extended WIT, control DCD)	Tissue ATP, ALT, liver function (with indocyanine green clearance test), oxygen uptake rate, bile production, targeted metabolomics (cofactors: ATP/ADP/AMP, NADH/NAD+, NADPH/NADP, FAD and GSH/GSSG) and untargeted metabolomics analysis, histology	Luminescence-based assay (Cell Viability Kit; Biovision)	A significant 4.12 fold increase in ATP level was observed post SNMP. The absolute ATP level at the end of SNMP was lowest in DCD (WIT > 0.5 h) group, followed by steatotic DCD group and highest in DCD (WIT < 0.5 h) group. Oxygen consumption was highest in DCD (WIT > 0.5 h) group. From transmission electron microscopy of biopsies from the three groups, mitochondrial injury score was highest in DCD (WIT > 0.5) group, with increased membrane permeability and swelling observed. Mitochondrial scores were suggested to be negatively associated with absolute ATP levels post SNMP. In this study, metabolomic analyses of livers with steatosis and prolonged WIT were conducted, suggesting that differences in metabolic factors and perfusion parameters may be closely linked to ATP recovery in livers.
Ferrigno [27]	28	0.5	6	Oxygenated Krebs-Henseleit (KH) medium with glucose, calcium chloride, with or without Ringer Lactate	6	SNMP, graft viability assessed by NMP reperfusion (2 h)	20	Rat, livers (DCD & 2 models of fatty livers: MCD diet & obese Zucker fa/fa)	Tissue ATP, ADP, AST, ALT, LDH, total bile production, bile flow, biliary enzymes, fatty acid evaluation, total lipids	Bioluminescence assay kit CLS II (Roche Molecular Biochemicals, Milan, Italy)	The effects of SNMP followed by NMP viability assessment vs. SCS on ATP/ADP ratio recovery were compared in a DCD liver model and two fatty liver models. A combined method of OW + CS was also compared to SCS and to SNMP in DCD livers. Higher ATP/ADP ratio and reduction of hepatic injury markers were reported in OW+CS group vs. SCS-only. Interestingly, comparable ATP/ADP ratios were reported in OW+CS (4 °C) and SNMP group. In the two fatty liver models, increase in ATP/ADP ratio was reported in SNMP-treated obese Zucker livers vs. SCS, but not in SNMP-treated MCD livers. The study suggested that preservation temperature and dynamic MP may not be the only modalities for graft resuscitation, but an oxygen washout prior to SCS at 4 °C might also facilitate ATP recovery in DCD livers. This is less clear in fatty livers.

* oxygenated unless otherwise stated ADP: Adenosine DiPhosphate; ALP: Alkaline phosphatase; ALT: Alanine Aminotransferase; AMP: Adenosine MonoPhosphate; AST: Aspartate Transaminase; ATP: Adenosine TriPhosphate; CIT: Cold Ischaemia Time; DBD: Donation after Brain Death; DCD: Donation after Circulatory Death; FAD: Flavin Adenine Dinucleotide; GDH: glutamate dehydrogenase; GGT: gamma-GT; GSH: Glutathione; GSSG: Glutathione disulfide; LDH: Lactate De-Hydrogenase; HMP: Hypothermic Machine Perfusion; MCD: Methionine-Choline Deficient Diet; MP: Machine Perfusion; NAD: Nicotinamide Adenine Dinucleotide; NADH: Nicotinamide Adenine Dinucleotide Hydrogen; NADP: Nicotinamide Adenine Dinucleotide Phosphate; NADPH: Nicotinamide Adenine Dinucleotide Phosphate Hydrogen. NMP: Normothermic Machine perfusion; OW: Oxygen Washout; SCS: Static Cold Storage; SNMP: Subnormothermic Machine Perfusion; WE: Williams medium E; WIT: Warm Ischaemia Time.

**Table 3 jcm-08-01421-t003:** Characteristics and outcomes of key pre-clinical and animal studies investigating COR preservation of livers.

Study	N	WIT, h	CIT, h	Perfusate	Perfusion Time, h	Perfusion Technique *	Temp, °C	Model	End Points	ATP Measurement Technique	Study Outcome
Minor [18]	12	-	18	CN with low potassium, mannitol, ketoglutarate, histidine, tryptophan	1.5	COR, graft integrity assessed by ex vivo NMP reperfusion	Slow gradual increase of perfusate temperature from 8 to 20 °C	Porcine, livers	Tissue ATP, ADP, AMP, ALT, AST, TNF-α, lipid peroxidation (oxygen free radical induced tissue injury), vascular resistance, bile production, caspase 3, Atg6, histology	Enzymatically determined in neutralized supernatant for protein extraction with perchloric acid of freeze dried tissue samples with hexokinase & glucose-6-phosphate dehydrogenase reactions for ATP conversion.	COR, HMP and SNP were tested in this model, with SCS as control. Tissue ATP levels significantly increased in COR group and SNP group, compared to HMP group (*p* < 0.05). Very low tissue ATP level was detected in SCS group. Restoration of end-ischaemic tissue energetics was comparable between COR and SNP, and COR appeared to be more protective of IRI and function, with reduction in TNF-α expression, caspase-3 activation, histology injury score and increase in bile production and portal vascular perfusion resistance.
von Horn [28]	18	0.5	18	AQIX RS-I Solution (serum and animal/human protein free), stable buffer (pH 7.20–7.45)	1.5	COR, graft integrity assessed by ex vivo NMP reperfusion	2 test groups: COR20 & COR35 (gradual increase up to 20 °C or 35 °C)	Rat, DCD livers	Tissue ATP, AST, ALT, blood gas analysis (pH, glucose, lactate), bile production, histology	ATP commercial test kit (Abcam, Cambridge, UK)	Tissue ATP levels in COR20 group and COR35 group were comparable (2.56 +/− 0.32 vs. 2.44 +/−0.27 μmol/g/ dw) and higher than that in SCS control group (1.79 +/− 0.07 μmol/g/ dw). No significant differences in energetic recovery, ALT release (reduced in both test groups), histopathology and increase in bile flow were observed for both test groups. Bile production was measurable in COR35 group prior to reperfusion, compared to inconsistent bile production in COR20 group. Further studies are required to differentiate whether COR35 offers greater improvement in function, viability and IRI compared to COR20.

* oxygenated unless otherwise stated. ADP: Adenosine DiPhosphate; ALT: Alanine Aminotransferase; AMP: Adenosine MonoPhosphate; AST: Aspartate Transaminase; Atg6: Autophagy Related Protein 6; ATP: Adenosine TriPhosphate; CIT: Cold Ischaemia Time; CN: Custodiol-N HTK solution; COR: Controlled Oxygenated Rewarming; DCD: Donation after Circulatory Death; IRI: Ischaemia-Reperfusion Injury; NMP: Normothermic Machine perfusion; TNF-alpha: Tumour Necrosis Factor-Alpha; WIT: Warm Ischaemia Time.

**Table 4 jcm-08-01421-t004:** Characteristics and outcomes of key pre-clinical and animal studies investigating NMP preservation of livers.

Study	N	WIT, h	CIT, h	Perfusate	Perfusion Time, h	Perfusion Technique	Temp, °C	Model	End points	ATP Measurement Technique	Study Outcome
Xu [29]	12	0 or 1	0 or 2	Whole blood with sterile porcine plasma, hydrocortisone, insulin, penicillin, streptomycin and heparin	4	NMP	39	Porcine, DCD livers	Tissue ATP, AST, ALT, ALP, bile production, blood gas analysis, histology	ATP Colorimetric Assay Kit (K354-100; Biovision Inc., Mountain View, CA)	After being subjected to extended WIT (1h) and CIT (2h), tissue ATP levels were restored to 80% of initial starting level in NMP group. There was improvement in hepatocyte necrosis post-NMP. The restoration of ATP and mitochondrial integrity were suggested to be the underlying reason of allograft viability and functionality in the NMP group.
Maida [30]	33	0.5	6 (after SOWP)	KH buffer (with or without PGE1) with glucose, sodium, potassium, tromethamine	0.5	SOWP before static cold storage, (with or without PGE1 addition)	37	Rat, DCD livers, with DBD group as positive control	Tissue ATP, ADP, ADP:ATP ratio, AST, ALT, ICAM-1, HMGB-1, histology, malondialdehyde (MDA, measures intensity of oxidative stress), survival rate post-transplantation	High-performance liquid chromatography	This study was an orthotopic rat DCD liver transplant model which evaluated the effects of SOWP with or without the addition of PGE1 (which was shown to resuscitate mitochondrial function in previous studies) on survival and graft viability. Previous studies by the same group demonstrated that SOWP prior to cold preservation was effective in preventing warm IRI of DCD rat livers. In this study, a significant increase in ATP level (*p* = 0.01) and reduction in ADP/ATP ratio (*p* = 0.02) were detected in SOWP (+PGE1) group compared to SOWP group and SCS group. 4-week survival rate was 81% in SOWP (+PGE1) group vs. 61% (SOWP group) and 100% (DBD group, control).

ADP: Adenosine DiPhosphate; ALP: Alkaline phosphatase; ALT: Alanine Aminotransferase; AST: Aspartate Transaminase; ATP: Adenosine TriPhosphate; CIT: Cold Ischaemia Time; DBD: Donation after Brain Death; DCD: Donation after Circulatory Death; HMGB-1: High Mobility Group Box 1; ICAM-1: Intercellular Adhesion Molecule-1; KH: Krebs-Henseleit; PGE1: prostaglandin E1; NMP: Normothermic Machine Perfusion; SOWP: Short Oxygenated Warm Perfusion; WIT: Warm Ischaemia Time.

**Table 5 jcm-08-01421-t005:** Characteristics and outcomes of key pre-clinical and animal studies investigating the relationship of metabolism and different perfusion temperatures in the liver.

Study	N	WIT, h	CIT, h	Perfusate	Perfusion Time, h	Perfusion Technique	Temp, °C	Model	End Points	ATP Measurement Technique	Study Outcome
Ferrigno [31]	24	-	-	Modified Krebs-Henseleit buffer with calcium chloride, NAC	6	-	10 or 20 or 30 or 37	Rat, livers	Tissue ATP, LDH, oxygen uptake, bile production, glycogen, HIF-1a	ATPlite luminescence assay kit (Perkin Elmer Inc., Waltham, MA, United States)	Tissue ATP levels in subnormothermic group (20 °C) significantly higher than that in 30 °C group and 37 °C group (*p* < 0.01), and also higher than that in 10 °C group (*p* < 0.01). However, ATP level in 10 °C group was higher than that in the normothermic groups (30 °C and 37°C). There was no increase in HIF-1a mRNA expression (which measures tissue hypoxia) in hypothermic (10 °C) group and subnormothermic (20°C) group, compared to 30 °C group (*p* < 0.01) and 37 °C group. A decrease in bile flow was observed in the normothermic groups, compared to 10 °C group and 20 °C group. In accordance with previous literature, bile flow may be dependant on activity of ATP-driven pumps. Thus, a decrease in tissue ATP might reflect a decrease in bile formation and flow. MP of livers at 10 °C and 20 °C was suggested to be protective of IRI and anaerobiosis, compared to normothermic temperatures.

ATP: Adenosine TriPhosphate; CIT: Cold Ischaemia Time; HIF-1a: Hypoxia Inducible Factor; IRI: Ischaemia-Reperfusion Injury; LDH: Lactate De-Hydrogenase; mRNA: messenger RNA; NAC: N-Acetyl-Cysteine; WIT: Warm Ischaemia Time.

**Table 6 jcm-08-01421-t006:** Characteristics and outcomes of key pre-clinical and animal studies comparing HMP versus NMP or combined approach (HMP + NMP) preservations of liver.

Study	N	WIT, h	CIT, h	Perfusate	Perfusion Time, h	Perfusion Technique *	Temp, °C	Model	End Points	ATP Measurement Technique	Study Outcome
Schlegel [32]	25	0.5 or 1	4	NMP (either oxygenated diluted full blood or with a leukocyte and platelet depleted blood perfusate) with bicarbonate, prostacyclin, amoxicillin, HOPE (starch-free UW)	4	NMP or HOPE, grafts assessed by ex-vivo normothermic reperfusion and liver transplantation	37 °C (NMP), 4 °C (HOPE), 4 °C (SCS)	Rat, DCD livers	Tissue ATP, oxidative damage of DNA by reactive oxygen species, HMGB-1, TLR-4, IL-6, ENA 78, NSE, ICAM-1, TNF-α, AST, histology, bile production	UV spectroscopy (340 nm) with hexokinase and glucose-6-phosphate dehydrogenase	Higher ATP levels in HOPE groups (DCD with WIT = 0.5 or 1 h) compared to NMP groups (*p* ≤ 0.0001). ATP levels in both MP groups were higher than control groups, but HOPE offered more protection of mitochondrial function of cells than NMP. ATP levels were consistent with and corresponded to bile flow in MP and control groups. HOPE might be superior to NMP in a clinically relevant model in terms of protection of hepatocyte and non-parenchymal injury and survival rates in livers treated with WIT (0.5 h) [90%; 9/10 vs. 70%; 7/10] and in livers treated with WIT (1 h) [63%; 5/8 vs. 0%; 0/10].
Boteon [17]	10	-	~8	NMP (Hemopure complemented with human albumin solution and other supplements), HOPE (UW)	6	NMP only vs. combined HMP (2 h) + NMP (4 h)	37 °C (NMP), 10 °C (HOPE)	Human, discarded livers (DBD and DCD)	Tissue ATP, pH, pO2, pCO2, O2 saturation, lactate level, bile production, histology, oxidative injury (uncoupling protein 2, 4-hydroxynonenal), tissue inflammation (CD14, CD11b, VCAM-1)	ATP Bioluminescent Assay kit (FLAA, Sigma-Aldrich Inc., St. Louis, MO)	For the HMP+NMP group, a median 1.8 fold increase in tissue ATP with a gradual decline in oxygen uptake were reported during the HOPE phase. Comparable end-perfusion tissue ATP levels were reported in viable grafts of HMP+NMP group vs. NMP group. A significant difference in changes in ATP was noted between viable and non-viable NMP livers (2.5 vs. 1.1 fold, *p* = 0.05). The combined approach of HOPE and NMP perfusion may be more effective in promoting recovery of mitochondrial function (HOPE phase) and allowing viability assessment (NMP phase) than NMP alone.

* oxygenated unless otherwise stated. AST: Aspartate Transaminase; ATP: Adenosine TriPhosphate; CIT: Cold Ischaemia Time; DBD: Donation after Brain Death; DCD: Donation after Circulatory Death; ENA 78: Epithelial neutrophil-activating protein 78; HMGB-1: High mobility group box protein 1; HOPE: Oxygenated Hypothermic Machine Perfusion; ICAM-1: Intercellular adhesion molecule 1; IL-6: Interleukin 6; NMP: Normothermic Machine perfusion; NSE: Neuron specific enolase; pCO2: partial pressure of carbon dioxide; pO2: partial pressure of oxygen; SCS: Static Cold Storage; TLR-4: Toll-like receptor 4; TNF-α: tumor necrosis factor α; UW: University of Wisconsin Solution; VCAM-1: Vascular Cell Adhesion Molecule 1; WIT: Warm Ischaemia Time.

**Table 7 jcm-08-01421-t007:** Characteristics and outcomes of key pre-clinical and animal studies investigating HMP preservation of kidneys.

Study	N	WIT, h	CIT, h	Perfusate	Perfusion Time, h	Perfusion Technique *	Temp, °C	Model	End Points	ATP Measurement Technique	Study Outcome
Buchs [44]	7	0 or 0.5	0 or 4 or 8 or 18	KPS-1 with glucose, mannitol, glutathione (reduced form), CaCl_2_, NaOH, KH_2_PO_4_	8 or 8 & 18	HMP	2–4	Porcine, DCD	Tissue ATP, precursors (PME, NAD) after SCS or after 8 or 18 h of machine perfusion	^31^P NMR spectroscopy	HOPE appears to be effective in supporting ATP resynthesis in kidneys subjected to WIT, but is less effective in kidneys subjected to SCS. Immediate HOPE perfusion should be used after organ retrieval to maintain organ viability.
Kay [46]	18	Minimal (6.4 +/− 1.0 min)	2	AQIX or HOC with sodium, potassium, mannitol, citrate or UW with adenosine	6	NMP vs. HMP (control: SCS), ex vivo function testing by perfusion (37 °C) with autologous blood	30 °C (AQIX), 4 °C (HOC), 4 °C (UW)	Porcine, kidneys	Tissue ADP:ATP ratio (ATP and ADP), perfusate flow rate, histology, serum creatinine, creatinine clearance, urine output	Bioluminescence adenylate nucleotide ratio assay kit (Cambrex BioScience, Berkshire, UK)	Pre-perfusion ADP:ATP ratios were high, demonstrating effects of reduction in cell metabolism due to cold storage. Decrease in ADP:ATP ratios was reported in all NMP (with AQIX), HMP (with HOC) and SCS (with UW) groups, with comparable ADP:ATP ratios amongst the three groups.
Lazeyras [45]	10	-	10 (only 1 graft was subjected to cold storage)	KPS-1 with insulin, phosphate, HES	8	HMP	4	Porcine, DCD	Tissue ATP (n = 9 under O_2_ + HPP, n = 1 mimics cold-ischaemia injured kidney which is perfused, cold stored and reperfused), PME, NAD(H)	^31^P MR spectroscopy and ^31^P CSI	The combined methods enabled detection of ATP in grafts perfused under 100 kPa pO2. In the CI-injured kidney model (n = 1), ATP levels were detected in after initial perfusion, but not detected after 10 h of SCS. Upon reperfusion, ATP was detected again, reaching close to pre-ischaemic level. The ATP level is one of the important markers for cell and graft viability and this technique could be transferable to clinical evaluation of marginal donor kidneys.
Ravaioli [47]	20	-	>=20	Celsior with glutathione, NaOH, mannitol, low K^+^	3	PE-HMP vs. PE-O_2_ vs. PE	4	Human, discarded kidneys (DBD)	Tissue ATP, histology, pH, lactate, pO2, pCO2, RNA (HIF-1α, eNOS, β-actin, β-2microglobulin)	ATP determination kit (Cat. N A22066, Thermo Fisher, Waltham, MA, USA)	HMP (hyperbaric or normobaric) was superior to conventional SCS or unoxygenated MP in the net increase in graft tissue ATP with respect to baseline level. Limitations of study include the lack of a transplantation model post-MP for assessment of graft function and survival.
Kaminski [48]	15	1	-	KPS-1	20	HMP	8.2+/−1.0	Porcine, WIT- injured grafts	Cortical tissue ATP, histology, oxygen arterio-venous difference	ATP Bioluminescent Assay kit (Sigma)	Renal cortical ATP was restored in HMP-treated warm-ischaemia injured kidney grafts to control (non-ischaemic) group (3.3 mmol/L tissue). Tissue ATP in HMP group was significant higher than that in SCS group (5.8 vs. 0.06 mmol/L, *p* < 0.01). Measurement of tissue oxygen and tissue ATP could be added to the current restricted panel of clinical evaluation criteria of grafts.

* oxygenated unless otherwise stated. ADP: Adenosine DiPhosphate; ATP: Adenosine TriPhosphate; CaCl_2_: Calcium chloride; CIT: Cold Ischaemia Time; CSI: Chemical shift imaging; DBD: Donation after Brain Death; DCD: Donation after Circulatory Death; eNOS: nitric oxide synthase 3; HES: hydroxyethyl starch; HIF-1α: Hypoxia-inducible factor 1-alpha; HMP: Hypothermic Machine Perfusion; HOC: Hyperosmolar Citrate; HOPE: Oxygenated Hypothermic Machine Perfusion; KH_2_PO_4_ : Potassium Dihydrogen phosphate; K^+^: potassium; KPS-1: Kidney Perfusion Solution-1; NAD(H): Nicotinamide Adenine Dinucleotide (Hydrogen); NaOH: Sodium hydroxide; NAD: Nicotinamide Adenine Dinucleotide; pCO2: partial pressure of carbon dioxide; PE: Hypothermic Perfusion PE-HMP: hypothermic perfusion in hyperbaric oxygen; PE-O_2_: Hypothermic Oxygenated Perfusion; PME: Phosphomonoester; pO2: partial pressure of oxygen; SCS: Static Cold Storage; WIT: Warm Ischaemia Time.

**Table 8 jcm-08-01421-t008:** Characteristics and outcomes of key pre-clinical study investigating HMP preservation of pancreas

Study	N	WIT, h	CIT, h	Perfusate	Perfusion Time, h	Perfusion Technique *	Temp, °C	Model	End points	ATP Measurement Technique	Study Outcome
Leemkuil [57]	20	18 (DCD only)	~10	UW	6	HMP	4 to 7	Human, discarded pancreata (n = 10 DBD and n = 10 DCD)	Tissue ATP, tissue edema (wet to dry weight ratio), TBARS, amylase, lipase, LDH, islet isolation procedure, perfusion flow	ATP Bioluminescence assay kit CLS II (Roche Diagnostics GmbH, Boehringer Mannheim, Germany)	HMP preservation of DBD and DCD pancreata was associated with a 6.8 fold increase and 2.6 fold increase in ATP concentration respectively, with viable islets isolated from 2 samples and no reported edema formation and indications of tissue injury. This first report of dual arterial HOPE for human donor pancreas suggested that HOPE may help to improve graft viability.

* oxygenated unless otherwise stated. ATP: Adenosine TriPhosphate; CIT: Cold Ischaemia Time; DBD: Donation after Brain Death; DCD: Donation after Circulatory Death; HMP: Hypothermic Machine Perfusion; HOPE: Oxygenated Hypothermic Machine Perfusion; LDH: Lactate De-Hydrogenase; TBARS: Thiobarbituric acid reactive substance; WIT: Warm Ischaemia Time.

**Table 9 jcm-08-01421-t009:** Characteristics and outcomes of key animal study investigating NMP preservation of pancreas.

Study	N	WIT, h	CIT, h	Perfusate	Perfusion Time, h	Perfusion Technique *	Temp, °C	Model	End points	ATP Measurement Technique	Study Outcome
Kumar [59]	13	Minimal (4 to 5 min)	~2	Soltran kidney perfusion fluid with potassium citrate, sodium citrate, mannitol, magnesium sulphate	3 h (control, ‘high pressure grafts’, 50 mmHg) or 4 h (test, ‘low pressure’ grafts, 20 mmHg)	NMP	37	Porcine, DCD pancreata	ATP Synthetase Complex V activity, blood gas analysis, standard electrolytes, glucose, amylase, insulin, Caspase 3, M30 CytoDEATH	Immunohistochemical staining of biopsied grafts (note activity of ATP synthetase was graded: Grade I (best), >95% of slide section staining positive for target enzyme), Grade II (>90 but ≤95%), Grade III (>85 but ≤90%)	ATP Synthetase Complex V is one of the enzymes essential for ATP production and metabolic functions, such as insulin secretion (ATP-dependent beta islets of Langerhans signalling pathways) and this is one of the markers for graft viability. In this ex vivo porcine pancreas perfusion model, a significant reduction in amylase and increase in ATP Synthetase activity were reported in ‘low pressure’ test group vs ‘high pressure’ control group (*p* < 0.016). This physiological EVNPPP model may be useful for the study of whole pancreas preservation.

* oxygenated unless otherwise stated. ATP: Adenosine TriPhosphate; CIT: Cold Ischaemia Time; DCD: Donation after Circulatory Death; NMP: Normothermic Machine Perfusion; WIT: Warm Ischaemia Time.

**Table 10 jcm-08-01421-t010:** Preclinical and animal MP studies that reported ATP or ATP-related parameters as one of the end points (2009–2019).

Perfusion Technique	Organ	Study
HMP	Kidneys	Buchs (porcine) [44] Lazeyras (porcine) [45] Ravaioli (human, discarded kidneys) [47] Kaminski (porcine) [48]
	Liver	Westerkamp (human, discarded livers) [21]
	Pancreas	Leemkuil (human, discarded pancreas) [57]
Combined HMP+NMP	Liver	Boteon (human, discarded livers) [17]
HMP vs. NMP	Liver	Schlegel (rat) [32]
HMP vs. NMP	Kidneys	Kay (porcine) [46]
SNMP	Liver	Ferrigno (rat) [22] Berendsen (rat) [23] Bruinsma (rat) [24] Bruinsma (human, discarded livers) [25] Bruinsma (human, discarded livers) [26] Okamura (rat) [39] Ferrigno (rat) [27]
COR (controlled oxygenated rewarming)	Liver	Minor (porcine) [18] von Horn (rat) [28]
NMP	Liver	Xu (porcine) [29] Maida (rat) [30]
NMP	Pancreas	Kumar (porcine) [59]
SCS vs. HMP vs. SNMP vs. NMP	Liver	Ferrigno (rat) [31]

ATP: Adenosine TriPhosphate; COR: Controlled oxygenated rewarming; HMP: Hypothermic Machine Perfusion; MP: Machine Perfusion; NMP: Normothermic Machine Perfusion; SCS: Static Cold Storage; SNMP: Subnormothermic Machine Perfusion.

**Table 11 jcm-08-01421-t011:** MP studies that reported liver tissue ATP (2009–2019).

Temp/°C	Increase in Liver Tissue ATP Levels Post-MP vs. pre-MP Perfusion	Study Type
10–12	Westerkamp [21]	Human, DCD, DBD
20	Ferrigno [22]	Rat, DCD
	Berendsen [23]	Rat, DCD
	Bruinsma [24]	Rat
	Minor (COR20) [18]	Porcine
	Bruinsma [25]	Human, DBD, DCD
	Ferrigno (20 > 10 > 30 > 37 °C) [24]	Rat
	Bruinsma (DBD/DCD/ steatotic livers) [25]	Human
	von Horn (COR 20 °C > COR 35 °C) [28]	Rat, DCD
	Okamura [39]	Rat, steatotic livers
	Ferrigno [27]	Rat, DCD, steatotic livers
37	Schlegel [32] (Higher vs. SCS, but HOPE > NMP) **Study compares HOPE (4 °C) to NMP (37 °C) & SCS	Rat, DCD
	Maida [30] (short oxygenated rewarming prior to SCS)	Rat, DCD
39	Xu [29]	Porcine, DCD
Combined HMP (4 °C) & NMP (37 °C)	Boteon [17]	Human, DBD, DCD

ATP: Adenosine TriPhosphate; COR20: Controlled Oxygenated Rewarming at 20 °C; DBD: Donation after Brain Death; DCD: Donation after Circulatory Death; HMP: Hypothermic Machine Perfusion; HOPE: Oxygenated Hypothermic Machine Perfusion; MP: Machine Perfusion; NMP: Normothermic Machine Perfusion; SCS: Static Cold Storage.

**Table 12 jcm-08-01421-t012:** MP studies that reported tissue ATP in kidneys (2009–2019).

Temp/°C	Increase in Renal Tissue ATP Levels Post-MP vs. Pre-MP Perfusion	Study Type
4	Buchs [44]	Porcine, DCD
	Lazeyras [45]	Porcine, DCD
	Ravaioli [47]	Human, DBD
8	Kaminski [48]	Porcine
HOPE (4 °C) vs. NMP (30 °C)	Kay [46] (*comparable ATP in HOPE, NMP and SCS groups)	Porcine, WIT-injured

ATP: Adenosine TriPhosphate; DBD: Donation after Brain Death; DCD: Donation after Circulatory Death; HOPE: Hypothermic Oxygenated Machine Perfusion MP: Machine Perfusion; NMP: Normothermic Machine Perfusion.

**Table 13 jcm-08-01421-t013:** MP studies that reported tissue ATP in pancreas (2009–2019).

Temp/ °C	Increase in Pancreatic Tissue ATP Levels Post-MP vs. Pre-MP Perfusion	Study Type
4–7	Leemkuil [57]	Human (DBD, DCD)
37	Kumar [59]	Porcine DCD

ATP: Adenosine TriPhosphate; DBD: Donation after Brain Death; DCD: Donation after Circulatory Death; MP: Machine Perfusion.

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
