# Peer review of "The Effect of Preservation Temperature on Liver, Kidney, and Pancreas Tissue ATP in Animal and Preclinical Human Models"

_jcm, 2019, doi:10.3390/jcm8091421_

Round 1

Reviewer 1 Report

Organ preservation and transplantation have been in research focus for many years. The manuscript is describing the effect of the preservation temperature on liver, kidney, and pancreas ATP level and how this affects the outcome. However, I find it very difficult to do such a comparison between different studies without mentioning the composition of the preservation solution (as well as the duration, which you touched in some of the reported studies) which definitely affects the outcome and not only the temperature as the impression here.  

I think this review will be more comprehensive if the aforementioned details were added.

The tables are hazy and unclear. I couldn't read them. Please consider improving them.

Thanks

Author Response

Dear Reviewers,

We are most grateful for your review of the manuscript and your helpful comments.

We have updated the tables with the perfusion composition, the duration of the machine perfusion and also have improved their resolution for a better readability.

We are at your disposal should you have any further questions.

Reviewer 2 Report

Organ transplantation is still the only therapy for terminal and irreversible organ failure. Since the shortage of organs for transplant, the improvement of existing preservation techniques is required. The authors presented here a very good overview of the actual preservation methodologies for liver, kidney and pancreas transplant, taking in account ATP production as a energy-status marker for recovery. The work is clearly explained and I really appreciate the comparison between the different techniques and the tables are well articulated. Also the images are effective.

However, I suggest some editing corrections within the manuscript:

Please, uniform the uppercase or lowercase when you write the abbreviation in the extended form. Here, I report only some examples:

in the abstract line 10: “Machine Perfusion (MP)”

in the introduction line 26 “machine perfusion (MP)”

in the introduction line 28 “randomized clinical trials (RCTs)”

in the introduction line 30 “Donation after Brain Death (DBD)”

in the introduction line 34 “delayed graft function (DGF)”.

Please, uniform “ex-vivo” (lines 43 and 205) and “ex vivo” (lines 105, 250, 454 and 457).

Please, uniform the typing font for lines 43-45, 50 “(Small interfering RNA)”, 403-407, 429-431.

Please, uniform the typing font for table captions.

Please, correct “(fig 1)” in line 120 in “(fig.1).

Please, delete a brackets in line 141.

Please, correct “et al” in “et al.” in lines 190, 271, 288, 356, 369, 394.

Please, correct the extended form of RCTs: in line 28 RCTs stays for “randomised clinical trials”, while in line 228 it stays for “Randomized Controlled Trial” (without the final “s”), as in the list of abbreviations.

Please, correct the “°C” symbols in line 249.

Please, add a space in lines 316, 330, 376, 404, 421.

Moreover, the concepts of the tables are clear, but at the moment the tables appear blurred (maybe for poor resolution of draft manuscript).

My opinion is “Accepted, with minor revision”.

Author Response

Dear Reviewers,

We are most grateful for your review of the manuscript and your helpful comments.

We have edited the text as per your suggestion and updated the tables with an improved resolution for a better readability.

We are at your disposal should you have any further questions.